# Antioxidative Properties of Machine-Drip Tea Prepared with Citrus Fruit Peels Are Affected by the Type of Fruit and Drying Method

**DOI:** 10.3390/foods11142094

**Published:** 2022-07-14

**Authors:** Beom-Gyun Jeong, Yu-Jeong Gwak, Jeong Kim, Won-Ho Hong, Su-Jin Park, Md. Atiqual Islam, Jiyoung Jung, Jiyeon Chun

**Affiliations:** 1Department of Food Science and Technology, Sunchon National University, Suncheon 57922, Korea; fusionchef@nate.com (B.-G.J.); plvnwh3@naver.com (Y.-J.G.); rlawjd030277@naver.com (J.K.); dnjsgh0531@naver.com (W.-H.H.); bbakssen@naver.com (S.-J.P.); atiquali@gmail.com (M.A.I.); jypw47@gmail.com (J.J.); 2Suncheon Research Center for Bio Health Care, Suncheon 57922, Korea; 3Kimchi Science and Industrialization Institute, Sunchon National University, Suncheon 57922, Korea

**Keywords:** citrus peel, air-drying, freeze-drying, machine-drip tea, antioxidant activity, flavonoids

## Abstract

Citrus peels are generally discarded as waste, although they are rich sources of health-promoting compounds. This study investigated the properties of citrus peels for development as a potential functional tea ingredient. Three citrus peel powders (DCPPs, Cheonhyehyang, Hallabong, and orange) which were dried by air- and freeze-drying, were used to prepare machine-drip tea. Then, total polyphenol compounds (TPCs), flavonoids, and the DPPH radical scavenging activity of DCPPs and teas were evaluated. Freeze-dried DCPPs had relatively higher TPC s (16.47–21.11 mg GAE/g) and DPPH radical scavenging activity (3.25–16.43 mg GAE/g) than air-dried DCPPs; TPCs (14.06–19.12 mg GAE/g) and DPPH radical scavenging activity (1.80–3.22 mg GAE/g). In contrast, air-dried DCPPs were more effective in machine-drip teas, showing a higher range of TPCs (50.64–85.12 mg GAE/100 mL) and DPPH radical scavenging activity (1.05–3.86 mg GAE/100 mL) than freeze-dried DCPPs; TPCs (40.44–46.69 mg GAE/100 mL) and DPPH radical scavenging activity (0.56–1.08 mg GAE/100 mL). Among citrus varieties, Cheonhyehyang had the highest TPCs and DPPH radical scavenging activity in both DCPP and tea. Four flavonoids (Hesperidin, Naringin, Nobiletin, and Tangeretin) mainly existed in citrus peels. The amount of hesperidin was highest; therefore, Hallabong and orange exhibited higher total flavonoid contents. However, freeze-dried Cheonhyehyang peel and air-dried Cheonhyehyang tea, which showed the highest TPCs and DPPH radical scavenging activity, had higher nobiletin and tangeretin. This implies that nobiletin and tangeretin strongly influenced the antioxidant activity of citrus peels with TPC. This research provides essential information for the tea industry looking for functional ingredients. In addition, it helps to reduce by-products by using citrus peel powders.

## 1. Introduction

Healthy living and disease prevention are of major interest as life expectancy increases. Thus, the development of new functional food sources and products for daily consumption is gaining attention, in addition to specialized nutrient supplements. Most research focuses on investigating the antioxidative properties of food materials that can alleviate oxidative stress in the body caused by reactive oxygen species (ROS). ROS are highly reactive free radical chemical forms that cover both oxygen-containing radicals and certain nonradical oxidizing agents. Oxygen radicals include superoxide (O_2_^−^), hydroxyl (OH^−^), peroxyl (RO_2_^−^), and hydroperoxyl (HO_2_^−^) radicals, and nonradical oxidizing agents are hydrogen peroxide (H_2_O_2_), hypochlorous acid (HOCI), and ozone (O_3_) [1]. Excessive generation of ROS can cause oxidative damage to DNA and cell membranes, eventually inducing various chronic diseases in the human body.

Phenolic compounds, including flavonoids and polyphenols, widely distributed in plants, are commonly studied as health-promoting bioactive compounds and dietary components. Their various health-promoting properties, such as antioxidant and antimicrobial activity and the prevention of cardiovascular disease and brain aging, have been demonstrated in previous studies [2,3,4]. Thus, edible plants rich in phenolic compounds are generally targeted in the food industry to develop new high-value products with strengthened functionality [5,6]. The antioxidant activity of Jinyangju, a traditional Korean rice wine, increased with the addition of *Citrus junos* [5]. Jeong et al. [6] investigated the changes in polyphenol contents in buckwheat tea by conducting various manufacturing processes to investigate the most efficient condition for achieving health benefits.

Currently, coffee consumption is proliferating, and drinking coffee between meals has become a part of daily life in Korea [7]. Meanwhile, particular interest in healthy functional tea products and the awareness of caffeine’s adverse effects in coffee has increased. Some herbal tea capsules are compatible with coffee machines available on the market, making it convenient to produce a variety of healthy functional tea products. Accordingly, people can easily access health-promoting bioactive compounds by consuming tea in their diets. Various factors should be considered to meet the market demand for good-quality tea capsules in taste and functionality.

Citrus fruit is one of the most widely consumed fruits worldwide and contains numerous bioactive compounds such as dietary fiber, L-ascorbic acid, phenolic acid, flavonoids, limonoids, and carotenoids [8,9]. Moreover, the citrus peel contains higher phenolic acid and flavonoid contents, with considerably higher antioxidant activity than other parts of citrus fruit [10,11,12,13]. Naringin, hesperidin, nobiletin, and tangeretin are the main citrus flavonoids. Naringin and hesperidin are the most prevalent flavanone glycosides, whereas nobiletin and tangeretin are two common polymethoxylated flavones (PMFs), which exist as O-methylated aglycones. Among them, nobiletin and tangeretine have been known to exhibit various health benefit functions and excellent antioxidant agents [8,10,14]. However, citrus fruit parts are generally consumed as a food source, whereas peels are discarded as waste. Traditionally, dried tangerine peel, referred to as ‘*Chen-pi*’, has been used as a folk medicine in Korea and China for expelling excessive phlegm with coughing, indicating the health benefits of citrus peel are well recognized. However, studies have mainly focused on the functionality and utilization of citrus peels of yuzu and tangerine [11,15]. There is only limited information on the peels of hybrid citrus species. The most consumed citrus fruits in Korea are the tangerine (*Citrus unshiu*), lime (*Citrus aurantifolia*), mandarine (*Citrus reticulata*), citron (*Citrus medica*), pomelo (*Citrus maxima*), and yuzu (*Citrus aurantium*). These days, hybrid citrus species such as grapefruit (*Citrus* × *paradisi*), Hallabong ((*C. unshiu* × *Citrus sinensis*) × *C. reticulata*), orange (*Citrus* × *sinensis*), and Cheonhyehyang ((*C. unshiu* × *C. sinensis*) × *C. reticulata*) × (*C. reticulata* × *C. sinensis*)) are becoming popular. Some citrus fruit peels, such as tangerine and yuzu, have historically been used for medicine and as functional foods; some studies have investigated the potential use of citrus peels as value-added products with abundant bioactive compounds [13], and as a source of biofuels [16]. In this study, three citrus peels (Cheonhyehyang, Hallabong, and orange) were dried in air (50 °C) or by freeze-drying and used in the form of dried citrus peel powder (DCPP) for preparing machine-drip tea. This study investigated the effects of drying methods and citrus fruit types on the antioxidative properties of citrus peels and machine-drip tea.

## 2. Materials and Methods

### 2.1. Chemicals and Reagents

Gallic acid, naringin, hesperidin, nobiletin, tangeretin, Folin–Ciocalteu’s phenol reagent, and 1,1-diphenyl-2-picrylhydrazyl (DPPH) were purchased from Sigma-Aldrich Chemical Co. (St. Louis, MO, USA). All other chemicals and solvents used were of analytical grade.

### 2.2. Preparation of Dried Citrus Peel Powder (DCPP) and Drip Tea

Cheonhyehyang (Jeju), Hallabong (Jeju), and orange (imported from California, USA) were purchased from a produce distribution center (Jinju, Gyeongnam, Korea). First, citrus fruits were washed, water-drained, and then peeled. The separated peel was sliced to 0.5 cm in width and then dried in air at 50 °C or freeze-dried at −70 °C. Air-drying was conducted using an air dryer (HB-501M, Hanbaek, Bucheon, Korea) at 50 °C for 5–7 h. For freeze-drying, citrus peels were frozen at −70 °C for 3 h and dried using a freeze dryer (Lyoph-pride series, Ilshin, Incheon, Korea) for 48 h. The moisture content of the dried peels was 5% or less. The dried peels were ground using a grinder (Grinder CGC-4PC3BC, Cuisinart, East Windsor, NJ, USA) and sieved (No. 50, Chung Gye Sang Gong Co., Seoul, Korea) according to citrus type (Cheonhyehyang, Hallabong, orange). Each dried citrus peel powder (DCPP) was sealed and stored at −20 °C until further use. Machine-drip tea was prepared using a capsule coffee maker (Keuring, Waterbury, VT, USA). First, 5 g of DCPP was placed on a filter paper (85 × 75 mm, Hany, Namyangju, Korea) in a k-cup and dripped with 100 mL of water (72 ± 2 °C). All machine-drip tea samples were cooled and filtered prior to the assay.

### 2.3. Color Measurement

The color of DCPPs and their drip tea was measured using a colorimeter (CR-200, Minolta, Osaka, Japan) at a calibration of lightness (L): 97.10, redness (a): −0.17, and yellowness (b): 1.99 on a standard white plate. Each sample’s L*, a*, and b* values were measured in triplicate and are expressed as the mean ± standard deviation.

### 2.4. Total Polyphenol Content (TPC)

TPCs of DCPPs and drip tea were measured as described by Folin and Denis (1912) [17], with minor modifications. First, the polyphenols were extracted by incubating DCPP (0.5 g) with 10 mL of methanol (80%, *v*/*v*) in a water bath for 2 h at 60 °C. The extract samples were filtered through a 0.45 μm membrane filter (DISMIN^Ⓡ^-13CP, Toyo Roshi Kaisha, Ltd., Tokyo, Japan), and 100 μL of the filtered sample was mixed with 1 mL of deionized water. Then, 100 μL of 50% Folin–Ciocalteu’s phenol reagent and 200 μL of 5% sodium carbonate were added. The mixed solution was placed on a 96-well microplate (SPL Life Sciences Co., Ltd., Pocheon, Gyeonggi, Korea) and incubated in the dark for 1 h. Absorbance was then measured at 750 nm using a microplate reader (Eon, Biotek, St. Winooski, VT, USA). Methanol (80%, *v*/*v*) and water were used as blanks for DCPP and drip tea samples, respectively. The TPC was expressed as mg gallic acid equivalent (GAE)/g for DCPP and μg GAE/100 mL for drip tea.

### 2.5. Flavonoid Content (Naringin, Hesperidin, Nobiletin, and Tangeretin)

The flavonoid contents of DCPPs and drip tea samples were measured according to the method described by Xu et al. (2008) [9]. The flavonoids were extracted by incubating 0.2 g of DCPP in 8 mL of a solution containing methanol and dimethyl sulfoxide (1:1, *v*/*v*) for 12 h at room temperature. The extract samples were filtered through a 0.45 μm membrane filter (Tokyo Roshi Kaisha, Ltd., Tokyo, Japan) and subjected to HPLC-DAD (Agilent, Santa Clara, CA, USA) analysis. For the drip tea sample, cool-down drip tea (2 mL) was mixed with 6 mL of a mixture of methanol and dimethyl sulfoxide (1:1, *v*/*v*), subsequently incubated for 12 h at room temperature, filtered, and used for HPLC analysis. The flavonoids in sample extracts were separated using a Vydac 201TP C18 column (4.6 mm × 250 mm, 5 μm; Grace, Columbia, MD, USA). The A and B mobile phases were 0.1% glacial acetic acid in water and 0.1% glacial acetic acid in acetonitrile, respectively. Solvents A and B were run at a flow rate of 1.0 mL/min, using a gradient of 85% A (15% B) at 0 min, decreasing to 35% A for 35 min, decreasing to 0% A for 10 min, steady at 0% A for 5 min, and then increasing to 85% A for 5 min. A column was equilibrated with 85% A for 10 min before the next injection. Naringin and hesperidin were detected at 283 nm, and nobiletin and tangeretin were detected at 330 nm.

### 2.6. DPPH Radical Scavenging Activity

DPPH radical scavenging activity was determined according to the method described by Blois (1958) [18], with minor modifications. First, DCPP (0.5 g) was mixed with 10 mL of 80% (*v*/*v*) methanol and incubated in a water bath (HB-205 SW, Han-Baek Scientific Co., Bucheon, Korea) for 2 h at 60 °C. DCPP extract was filtered through a 0.45 μm membrane filter (Tokyo Roshi Kaisha, Ltd.) prior to use. An aliquot (200 μL) of the filtered extract was mixed with 800 μL of 0.2 mM DPPH solution and then kept in the dark for 30 min. Drip tea samples (200 μL) were also treated in the same way. The absorbance of DCPPs and drip tea samples was determined at 512 nm using a spectrophotometer. Methanol (80%) and deionized water were used as blanks for DCPPs and machine-drip tea, respectively. The DPPH radical scavenging activity was expressed as mg GAE/g for DCPP and μg GAE/100 mL for drip tea.

### 2.7. Statistical Analysis

All experiments were performed in triplicate. Statistical analyses were performed using the SPSS (SPSS 21.0, SPSS Inc., Chicago, IL, USA). One-way ANOVA was performed, and differences between means of samples were analyzed by *t*-tests and Duncan’s multiple range test (*p* < 0.05).

## 3. Results and Discussion

### 3.1. Characteristics of DCPP

#### 3.1.1. Color

Table 1 presents the Hunter L*, a*, and b* values of freeze-dried (−70 °C) and air-dried (50 °C) DCPPs (Cheonhyehyang, Hallabong, and orange). L*, a*, and b* values of DCPPs varied depending on the citrus fruit varieties and drying method. According to citrus varieties, Cheonhyehyang DCPP showed a relatively darker and reddish-orange color (lower L*-value and higher a* and b* values) than the other DCPPs. On the other hand, the Hallabong DCPP showed a light-yellow color, with the highest L* value and the lowest a* value among all DCPPs.

Furthermore, air-dried DCPPs showed significantly lower L* and b* values and higher a* values than freeze-dried DCPPs, indicating that air-drying resulted in a darker and deep reddish color. A lower L*-value has been reported in air-dried *Salicornia herbacea* [19] and *Codonopsis Ianceopate* Saengsik [20]. In addition, Gan et al. [21] reported that browning increased in mung bean sprouts after hot air-drying. Freeze-drying was conducted under a vacuum at a low temperature without exposure to oxygen; therefore, color, texture, and flavor changes were lesser than after hot air-drying. On the other hand, air-dried food can be thermally affected by exposure to high temperatures and oxygen [22]. Browning in color is generally caused by heat treatment in dehydrated foods.

The Cheonhyehyang DCPP showed the most significant differences in L*, a*, and b* values compared with other citrus peels, whereas orange DCPP had the minimum changes by drying methods (Table 1). It is presumed that Cheonhyehyang might be relatively more susceptible to temperature changes than orange and Hallabong DCPPs.

#### 3.1.2. Total Polyphenol Contents (TPCs)

The TPC is an important criterion to determine the potential health benefit value of plant-based foods because polyphenols have been reported to exert a variety of pharmacological activities and prevent degenerative diseases caused by excess ROS [2,4]. Figure 1A depicts the TPCs in freeze-dried (−70 °C) and air-dried (50 °C) DCPPs. Freeze-dried Cheonhyehyang and Hallabong DCPPs showed approximately 33% and 20% higher TPCs than air-dried DCPPs. Conversely, the air-dried orange DCPP had a 16% higher TPC than the freeze-dried orange peel. Thus, the freeze-dried Cheonhyehyang (21.11 mg GAE/g) had the highest TPC, and air-dried orange DCPP had the second highest TPC. In previous studies, freeze-dried Cheonhyehyang peel had a significantly higher TPC than Hallabong and navel orange peel [11,12]. The TPC of freeze-dried Cheonhyehyang peel was 33 mg GAE/g [12] and 30.2 mg GAE/g [11], which was relatively higher than this study’s TPC of freeze-dried Cheonhyehyang DCPP (21.11 mg GAE/g). The TPC can vary depending on the variety, harvest time, and standard substance used for the assay [23].

According to Jeong et al. [24], the TPC of *Eleutherococcus senticosus* significantly decreased with an increase in temperature from 40 °C to 80 °C during the air-drying process. Park et al. [25] reported that the TPC and antioxidative effect of air-dried (50 °C) Schizandra fruit was significantly lower than that of freeze-dried fruit, suggesting that some beneficial compounds were destroyed after exposure to hot temperature. In contrast, the TPC and antioxidant activity of citrus unshie peel increased as the heating temperature increased (50–150 °C) because of newly formed small molecular phenolic compounds during heat treatment. TPCs of air-dried mung bean sprouts increased with an increase in temperature from 40 °C to 80 °C [21]. The air-drying process changed the chemical compounds’ profile, including phenolic compounds, depending on temperature [21,26]. Therefore, it could be postulated that whatever phenolic compounds food have are critical to affecting TPCs after the drying process.

#### 3.1.3. Flavonoid Content (Naringin, Hesperidin, Nobiletin, and Tangeretin)

Table 1 depicts the main citrus flavonoid (naringin, hesperidin, nobiletin, and tangeretin) contents of DCPPs. Each flavonoid’s contents are varied based on the citrus variety leading to a highly different sum of four flavonoid contents. The sum of flavonoid contents in Cheonhyehyang DCPP was 457.4–567.1 μg/g, whereas Hallbong and orange DCPPs showed considerably higher contents with 3343.5–3522.4 μg/g and 3661.9–4489.2 μg/g, respectively (not shown). In Hallabong and orange DCPPs, the order of flavonoid contents was hesperidin > naringin > nobiletin > tangeretin, indicating that flavonoid glycosides were more predominant. This result agrees with a previous study where flavanone hesperidin was the most abundant polyphenol in citrus peels: navel orange, lemon, and clementine [13]. In contrast, in Cheonhyehyang DCPP, the order was nobiletin > hesperidin > tangeretin > naringin, indicating the proportion of PMFs (nobiletin, tangeretin) was relatively high compared with Hallabong and orange DCPPs. In addition, hesperidin was relatively small in Cheonhyehyang compared with other citrus peels, leading to the smallest total flavonoids; nevertheless, freeze-dried Cheonheyhyang peels exhibited the highest TPC (Table 1, Figure 1A). PMFs in Cheonhyehyang accounted for 79.0% (freeze-drying) and 78.0% (air-drying) of the total flavonoids. In contrast, the PMFs of Hallabong and orange were merely 7.9–8.5% and 2.9–3.1%, respectively. Cheonhyehyang might have a distinct phenolic compound profile from Hallabong and orange, influencing biological activities. It is well documented that nobiletin and tangeretine have various bioactive activities, such as antioxidant, anti-cancer, and anti-inflammatory properties [10,27,28].

All air-dried DCPPs showed similarly or 1.1–2.9-fold higher flavonoids than freeze-dried DCPPs (Table 1). According to flavonoids, nobiletin was not significantly different, and tangeretin differed merely by drying method for all citrus varieties. However, air-dried Cheonhyehyang DCPP had a more than 2-fold increase in naringin and hesperidin. Sung et al. [29] suggested that thermal processing at 150 °C elevated the flavonoid contents, especially in PMFs in mandarin peels. It has been reported that a high temperature of over 90 °C enhanced the release of phenolic compounds from softened cell wall polymers in citrus peels [30]. Our air-drying temperature (50 °C) was probably not high enough to strongly affect PMFs and TPCs. As seen in Figure 1A, Cheonhyehyang and Hallabong DCPPs showed 20–33% higher TPCs with freeze-drying treatment.

#### 3.1.4. DPPH Radical Scavenging Activity

Figure 1B displays the DPPH radical scavenging activity of DCPPs by the drying method. Freeze-dried DCPPs exhibited a significantly higher DPPH radical scavenging activity for all citrus varieties than air-dried DCPPs. In particular, freeze-dried Cheonhyehyang showed the most increased antioxidant activity (16.4 mg GAE/g), and the difference was approximately 80% by the drying method. Freeze-dried Hallabong and orange DCPPs showed approximately 39% and 45% higher antioxidant activities than air-dried DCPPs. In contrast, several studies stated that antioxidant activity increased with heat treatment, but the temperature was comparatively high: 50–150 °C in mandarin peel [31] and 150 °C for 50 min in *C. unshie* [29]. In contrast, Garau et al. [30] reported that the antioxidant activity of orange peel (*C. aurantium v. Canoneta*) decreased at over 60 °C drying temperature or for longer drying times at low temperature. The presumed antioxidant activity of citrus peel could vary depending on citrus varieties, temperature, and processing time.

It is well known that phenolic compounds are positively correlated with antioxidant activities because functional hydroxyl groups in phenolic compounds contribute to the antioxidant effect by scavenging radicals. The DPPH radical scavenging activity in this study was highly associated with TPC, excluding air-dried orange (Figure 1A,B). In addition, the DPPH radical scavenging activity appeared to be more closely associated with PMFs than total flavonoids, as seen in Figure 1 and Table 2.

## 4. Conclusions

Freeze-dried Cheonhyehyang DCPP, which had the highest DPPH radical scavenging activity, showed the lowest total flavonoids, with 79.0% of PMFs (nobiletin: 326.53 μg/g, tangeretin 35 μg/g). In contrast, the PMF proportions of orange and Hallabong DCPPs were 8.0–8.1% and 2.5–3.0%, respectively (Table 1). Thus, it could be postulated that nobiletin and tangeretin play a more crucial role in antioxidant activity than the total flavonoids. The phenyl benzopyrone group, a typical structure of PMFs, has a key role in antioxidant activity [32].

### 4.1. Characteristics of Machine-Drip Tea Prepared from DCPP

#### 4.1.1. Color of DCPP Drip Tea

Machine-drip tea was prepared using three DCPPs with a capsule coffee maker; Table 2 presents the Hunter L*, a*, and b* values of tea. Color is one of the main quality parameters in dehydrated foods because it influences properties and limits their potential application.

In this study, drip tea with air-dried DCPPs showed relatively lower L* and higher a* and b* values, indicating a darker red-orange than freeze-dried DCPP tea. It is supposed that the color of DCPPs influenced the drip-tea color by determining the same trend: lower L* and higher a* values (Table 1 and Table 2). In addition, similar to DCPP, Cheonhyehyang tea had more considerable differences in L* and a* values by drying method.

#### 4.1.2. TPC of DCPP Machine-Drip Tea

As seen in Figure 2A, air-dried DCPP tea showed significantly higher TPCs (47.63–85.12 mg GAE/100 mL) than freeze-dried DCPP teas (40.44–46.69 mg GAE/100 mL) for all citrus varieties. This result is in contrast to the TPC of DCPPs by the drying method. Freeze-dried DCPPs in Cheonhyehyang and Hallabong showed higher TPC than corresponding air-dried DCPPs (Figure 1A and Figure 2A). It was reported that air-drying caused more porous structures in *Salicornia herbacea* due to water evaporation and thermal damage on the surface than the freeze-drying method [19]. Therefore, it could be postulated that more TPC was transferred into tea through weakening the cell polymers of air-dried DCPPs.

The highest TPC was observed in air-dried Cheonhyehyang tea (85.12 mg GAE/100 mL), followed by air-dried Hallabong tea (55.61 mg GAE/100 mL) and air-dried orange tea (47.63 mg GAE/100 mL). In addition, Cheonhyehyang DCPP tea had about an 82% difference, whereas orange DCPP had just a 20% difference in TPC for the drying method. Cheonhyehyang peel was considered more susceptible to air-drying than other citrus peels.

#### 4.1.3. Naringin, Hesperidin, Nobiletin, and Tangeretin in DCPP Drip Tea

The flavonoid contents of drip tea prepared with DCPPs are presented in Table 2. Air-dried DCPP tea exhibited higher contents in naringin, nobiletin, and tangeretin than freeze-dried DCPP tea. In contrast, hesperidin showed higher contents in freeze-dried Hallabong and orange DCPP tea than in air-dried DCPP tea (Table 2).

In this study, naringin, one of the predominant flavonoids with hesperidin in Hallabong and orange DCPPs, was considerably low in tea. It led to a higher portion of tangeretin and nobiletin extraction in tea. A similar result was reported by Xu et al. [9], that the proportion of PMFs became considerable due to the low extraction ratio of hesperidin by hot water in Satsuma mandarin and Ponkan. In addition, they stated that a higher yield of PMFs was observed at lower water temperature as the temperature increased from 40 °C to 100 °C. Our machine-drip DCPP tea was prepared at approximately 72 °C.

The proportion of hesperidin was exceedingly high compared with other flavonoids; therefore, the sum of the four flavonoid contents was high in freeze-dried Hallabong, orange DCPP tea, and air-dried Cheonhyehyang DCPP tea. However, the PMF extraction was high in air-dried DCPP tea for all citrus varieties, with the order of Cheonhyehyang > Hallabong > orange tea. Peculiarly, PMFs accounted for 60.5% out of the sum of four flavonoids in air-dried Cheonhyehyang DCPP tea. PMFs are known to have various biological functions [8,10,14,33]. In addition, air-dried Cheonhyehyang DCPP tea had the highest TPC. Therefore, air-dried Cheonhyehyang DCPP might be more suitable as a citrus tea source.

#### 4.1.4. DPPH Radical Scavenging Activity of DCPP Drip Tea

The DPPH radical scavenging activities of drip teas prepared with DCPPs are presented in Figure 2B. Concerning the drying method, air-dried DCPP tea showed higher DPPH radical scavenging activity, although no significant difference was observed in Hallabong tea. In particular, air-dried Cheonhyehyang DCPP tea exhibited the most increased DPPH radical scavenging activity, with a substantial difference of 380% by drying method (air-dried tea: 3.86 mg GAE/100 mL, freeze-dried tea: 0.98 mg GAE/100 mL). Air-dried Cheonhyehyang tea showed remarkably higher TPC and PMFs (nobiletin, tangeretin) than other citrus varieties and freeze-dried Cheonhyehyang tea (Figure 2, Table 2). As supported by a previous study [8], these bioactive compounds probably lead to the highest antioxidant activity of air-dried Cheonhyehyang DCPP. Choi et al. [8] reported that nobiletin and tangeretin significantly suppressed nitric oxide (NO) production compared with the total flavonoid contents in citrus peel extracts. NO has been known to produce the potentially toxic oxidant peroxynitrite (ONOO−) by reacting with superoxide or singlet oxygen, causing significant damage to the cell structure in the body [33]. Moreover, nobiletin and tangeretine have been reported to prevent the initiation of lipid peroxidation by trapping free radicals to form stable phenoxy radicals [14].

This study determined the functional properties of DCPPs by freeze-drying and air-drying methods and citrus teas prepared using DCPPs. Freeze-drying was more effective in enhancing the TPC and DPPH radical scavenging activity of DCPPs, whereas teas prepared with air-dried DCPPs showed higher TPC and antioxidant activity. Mainly, Cheonhyehyang peel and tea had the highest levels in both TPC and antioxidant activity and TPC. Air-drying was more potent in DCPPs and teas in flavonoids, except for hesperidin of freeze-dried Hallabong and orange teas. Furthermore, the proportion of nobiletin and tangeretin was remarkably high in Cheonhyehyang compared with Hallabong and orange. This result implied that Cheonhyehyang peel rich in TPC, nobiletin, and tangeretin could be a functional food ingredient with a more substantial antioxidant activity in tea products.

## Figures and Tables

**Figure 1 foods-11-02094-f001:**
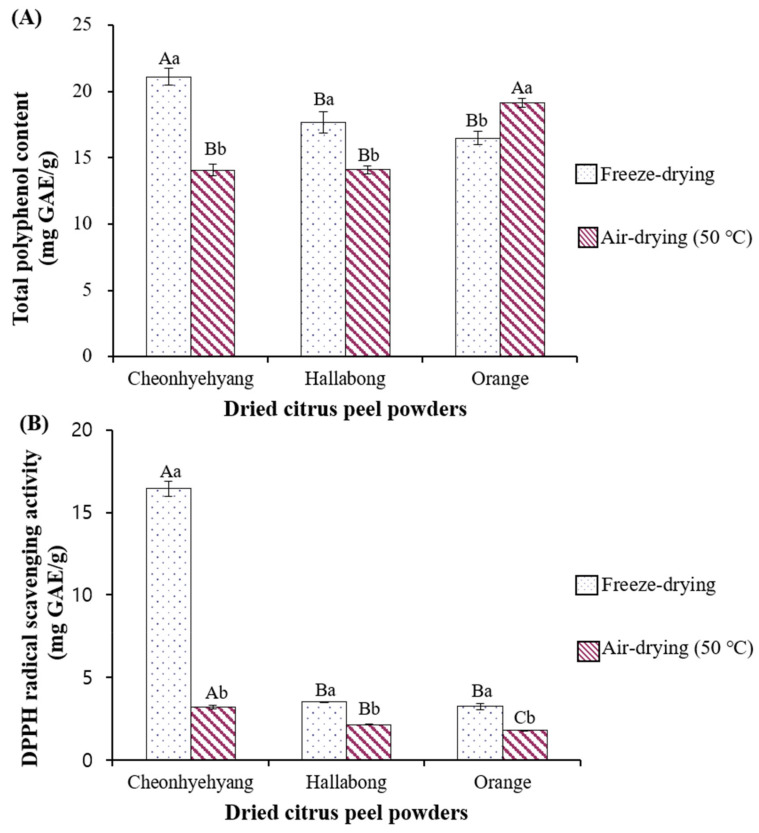
Total polyphenol contents and DPPH radical scavenging activity of DCPP prepared by freeze- and air-drying. (**A**) Total polyphenol contents, (**B**) DPPH radical scavenging activity). DCPP: dried citrus peel powder. Means with different capital letters on the same type bars (the same drying method) were significantly different by Duncan’s multiple range test at (*p* < 0.05, A > B). Means with different small letters on the same citrus fruit species were significantly different by *t*-test (*p* < 0.05, a > b).

**Figure 2 foods-11-02094-f002:**
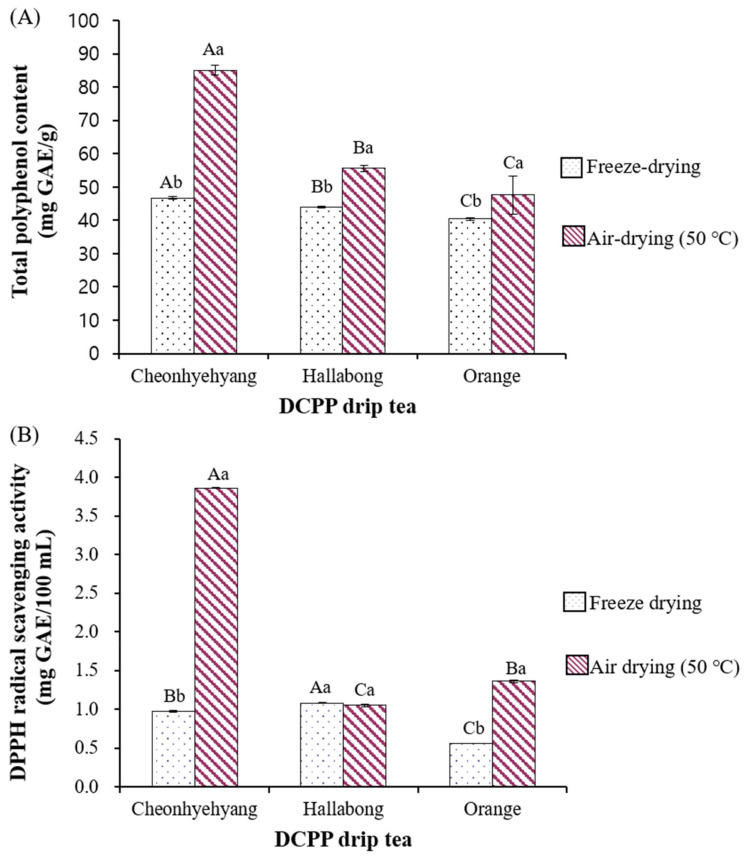
The total polyphenol contents and DPPH radical scavenging activity of machine-drip tea prepared from DCPP. (**A**) Total polyphenol contents, (**B**) DPPH radical scavenging activity). DCPP Dried citrus peel powder. Means with different capital letters in the same column for each color parameter were significantly different by Duncan’s multiple range test (*p* < 0.05, A > B > C). Means with different small letters on the same row were significantly different by *t*-test (*p* < 0.05, a > b).

**Table 1 foods-11-02094-t001:** Color and flavonoid contents of DCPPs prepared by freeze- and air-drying.

		DCPP ^1^	DCPP Drip Tea	
			Freeze-Drying	Air-Drying (50 °C)
Color	L	Cheonhyehyang	82.76 ± 0.03 ^B,a,2^	73.20 ± 0.02 ^C,b^
		Hallabong	87.14 ± 0.09 ^A,a^	80.75 ± 0.04 ^A,b^
		Orange	82.18 ± 0.05 ^C,a^	78.59 ± 0.05 ^B,b^
	a	Cheonhyehyang	1.67 ± 0.03 ^A,b^	8.28 ± 0.02 ^A,a^
		Hallabong	−4.85 ± 0.03 ^C,b^	0.37 ± 0.02 ^C,a^
		Orange	0.96 ± 0.02 ^B,b^	3.59 ± 0.02 ^B,a^
	b	Cheonhyehyang	83.33 ± 0.07 ^A,a^	70.15 ± 0.06 ^A,b^
		Hallabong	69.18 ± 0.14 ^B,a^	63.06 ± 0.08 ^B,b^
		Orange	53.22 ± 0.03 ^C,a^	52.24 ± 0.04 ^C,b^
Flavonoids (µg/g, dry basis)	Naringin	Cheonhyehyang	9.33 ± 3.12 ^C,b,2^	26.90 ± 6.15 ^C,a^
		Hallabong	660.56 ± 23.71 ^B,b^	714.73 ± 4.90 ^B,a^
		Orange	818.88 ± 41.94 ^A,b^	1002.65 ± 2.08 ^A,a^
	Hesperidin	Cheonhyehyang	86.57 ± 7.21 ^C,b^	183.21 ± 12.15 ^C,a^
		Hallabong	2416.92 ± 13.90 ^B,ns^	2523.38 ± 55.83 ^B^
		Orange	2734.75 ± 27.30 ^A,b^	3372.44 ± 22.01 ^A,a^
	Nobiletin	Cheonhyehyang	326.53 ± 2.87 ^A,ns^	321.10 ± 0.69 ^A^
		Hallabong	235.95 ± 3.88 ^B,ns^	250.54 ± 6.47 ^B^
		Orange	98.40 ± 1.75 ^C,ns^	102.20 ± 0.42 ^C^
	Tangeretin	Cheonhyehyang	35.00 ± 0.33 ^A,ns^	35.88 ± 0.24 ^A^
		Hallabong	30.05 ± 0.48 ^B,b^	33.75 ± 0.57 ^B,a^
		Orange	10.23 ± 0.03 ^C,b^	11.95 ± 0.02 ^C,a^

^1^ DCPP: dried citrus peel powder. ^2^ Means with different capital letters in the same column (the same drying method) were significantly different by Duncan’s multiple range test (*p* < 0.05, A > B > C). Means with different lowercase letters in the same row (the same citrus fruit species) were significantly different by *t*-test (*p* < 0.05, a > b). ns: not significant.

**Table 2 foods-11-02094-t002:** Color and flavonoid contents of machine-drip tea prepared from DCPP.

		DCPP ^1^	DCPP Drip Tea	
			Freeze-Drying	Air-Drying (50 °C)
Color	L	Cheonhyehyang	19.61 ± 0.64 ^A,a,2^	13.57 ± 0.39 ^C,b^
		Hallabong	17.73 ± 0.05 ^B,a^	14.41 ± 0.32 ^B,b^
		Orange	17.65 ± 0.03 ^B,a^	16.89 ± 0.08 ^A,b^
	a	Cheonhyehyang	−0.12 ± 0.06 ^B,b^	0.44 ± 0.09 ^A,B,a^
		Hallabong	0.30 ± 0.05 ^A,ns^	0.32 ± 0.05 ^B^
		Orange	0.34 ± 0.13 ^A,ns^	0.54 ± 0.10 ^A^
	b	Cheonhyehyang	3.14 ± 0.03 ^ns,b^	4.87 ± 0.10 ^A,a^
		Hallabong	3.14 ± 0.12 ^b^	4.11 ± 0.06 ^B,a^
		Orange	2.98 ± 0.08 ^b^	3.77 ± 0.03 ^C,a^
Flavonoids (µg/100 mL)	Naringin	Cheonhyehyang	18.59 ± 5.56 ^A,b,2^	36.54 ± 1.09 ^A,a^
		Hallabong	0.00 ± 0.00 ^C,b^	25.31 ± 2.19 ^B,a^
		Orange	8.97 ± 1.46 ^B,b^	15.11 ± 0.63 ^C,a^
	Hesperidin	Cheonhyehyang	185.24 ± 3.37 ^C,b^	597.94 ± 9.31 ^C,a^
		Hallabong	3547.54 ± 2.99 ^A,a^	2087.69 ± 26.29 ^A,b^
		Orange	2319.22 ± 3.24 ^B,a^	1124.25 ± 6.78 ^B,b^
	Nobiletin	Cheonhyehyang	335.93 ± 2.79 ^A,b^	900.29 ± 5.63 ^A,a^
		Hallabong	313.70 ± 1.49 ^B,b^	545.99 ± 6.55 ^B,a^
		Orange	162.80 ± 0.44 ^C,b^	201.84 ± 0.54 ^C,a^
	Tangeretin	Cheonhyehyang	29.69 ± 0.38 ^B,b^	75.41 ± 0.29 ^A,a^
		Hallabong	38.25 ± 2.01 ^A,b^	62.31 ± 0.42 ^B,a^
		Orange	14.02 ± 0.11 ^C,b^	16.74 ± 0.12 ^C,a^

^1^ DCPP: Dried citrus peel powder. ^2^ Means with different capital letters in the same column (the same drying method) were significantly different by Duncan’s multiple range test (*p* < 0.05, A > B > C). Means with different small letters on the same row (the same citrus fruit species) were significantly different by *t*-test (*p* < 0.05, a > b). ns: not significant.

## Data Availability

The data supporting the results of this study are included in the present article.

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
