# Peer review of "Antioxidative Properties of Machine-Drip Tea Prepared with Citrus Fruit Peels Are Affected by the Type of Fruit and Drying Method"

_foods, 2022, doi:10.3390/foods11142094_

Round 1

Reviewer 1 Report

The authors have studied on “Antioxidative properties of machine-drip tea prepared with cit- 2 rus fruit peels affected by the type of fruit and drying method”. The following are some of the comments in particular.

Mention the range of values of TPC and DPPH radical scavenging activity in the product in the abstract.

In Table 1-4. In ANOVA analysis the notations (superscripts) used are not explained properly.

Table 1-4 should be merged into one Table.

The same data presented in Table are presented in figures. Justify.

The References section has all old citations. The authors should include some currently published manuscripts.

Author Response

Mention the range of values of TPC and DPPH radical scavenging activity in the product in the abstract.

Information was added in the abstract

In Table 1-4. In ANOVA analysis the notations (superscripts) used are not explained properly.

It was corrected

Table 1-4 should be merged into one Table.

Among four tables, Table 1 (color) and Table 2 (flavonoids) were assigned for DCPPs, and Table 3 (color) and Table 4 (flavonoids) were for tea by DCPPs. Thus, we made two tables by merging table1 and 2 into table 1 for DCPP, and table3 and 4 became Table 2 for tea by DCPPs. That might be better for readers to follow the contents without misleading.

The same data presented in Table are presented in figures. Justify.

There were no same data between tables and figures

The References section has all old citations. The authors should include some currently published manuscripts.

New references were added.

Reviewer 2 Report

1) The section of "Introduction" needs to include more relavent and important references.

2)Information should be given about dried citrus peel powder (DCPP)

3) The section of "Results and discussions" should give in depth and the
application of the result of this study should be well discussed.

Author Response

1) The section of "Introduction" needs to include more relevant and important references.

More information and references were added in the introduction.

2) Information should be given about dried citrus peel powder (DCPP)

The information is added to the Materials and Methods (2.2).

3) The section of "Results and discussions" should give in depth and the
application of the result of this study should be well discussed.

Some Discussions were added.

Reviewer 3 Report

This study investigated the antioxidative contents and capacity of three dried citrus peel powders (DCPPs), air-dried or freeze-dried, in the production of machine-drip tea. This research was well performed and the results were clearly presented. I have some comments as follows:

1. Please cite the reference(s) for the first paragraph in the Introduction.

2. Page 1, Line 44: ROS also cover non-radicals such as H2O2. Please revise the definition of ROS.

3. Page 2, Line 89: The use of “,” can mislead the readers that citrus peels were dried in the air followed by freeze-drying. It should be presented as “were dried in air or by freeze-drying”.

4. Page 4, Line 165: The authors presented both results and discussion in Section 3, so it should be renamed into “Results and Discussion”.

5. The freeze-dried cheonhyehyang DCPP had the highest TCP (Fig. 1A) but the lowest flavonoid contents (Table 2). Please discuss the main polyphenols presented in this DCPP.

6. Why the air-dried cheonhyehyang DCPP had the lowest TCP (Fig. 1A) but the air-dried cheonhyehyang drip tea (Fig. 2A) had the highest TCP? 

Author Response

This study investigated the antioxidative contents and capacity of three dried citrus peel powders (DCPPs), air-dried or freeze-dried, in the production of machine-drip tea. This research was well performed and the results were clearly presented. I have some comments as follows:

  1. Please cite the reference(s) for the first paragraph in the Introduction.

Attached

  1. Page 1, Line 44: ROS also cover non-radicals such as H2O2. Please revise the definition of ROS.

Information added in line 49

  1. Page 2, Line 89: The use of “,” can mislead the readers that citrus peels were dried in the air followed by freeze-drying. It should be presented as “were dried in air or by freeze-drying”.

It was corrected
